Bacterial identification of the vaginal microbiota in Ecuadorian pregnant teenagers: an exploratory analysis

Salinas Ana María 1
Osorio Verónica Gabriela 1
Endara Pablo Francisco 1 2
Salazar Eduardo Ramiro 3
Vasco Gabriela Piedad 1 3
Vivero Sandra Guadalupe 3
Machado Antonio amachado@usfq.edu.ec machadobq@gmail.com 1
1 Instituto de Microbiología, Colegio de Ciencias Biológicas y Ambientales, Universidad San Francisco de Quito , Quito , Ecuador
2 Colegio de Ciencias de la Salud, Universidad San Francisco de Quito, Universidad San Francisco de Quito , Quito , Ecuador
3 Facultad de Ciencias Médicas, Universidad Central del Ecuador , Quito , Ecuador
Mastrolia Salvatore Andrea
Electronic publication date: 2018 Feb 21
Publication date: 2018
Volume: 6
Electronic Location ID: e4317
Received 2017 Aug 10; Accepted 2018 Jan 12
Copyright: ©2018 Salinas et al.
Copyright year: 2018
Copyright holder: Salinas et al.
License: This is an open access article distributed under the terms of the Creative Commons Attribution License, which permits unrestricted use, distribution, reproduction and adaptation in any medium and for any purpose provided that it is properly attributed. For attribution, the original author(s), title, publication source (PeerJ) and either DOI or URL of the article must be cited.
License URL: https://creativecommons.org/licenses/by/4.0/

Keywords: Lactobacillus sp., Pregnancy, Species identification, Vaginal microbiota, Bacterial vaginosis, 16S rRNA, Teenagers, Ecuador, 23S rRNA, Anaerobes

Funding: Proyectos Semilla from Universidad Central del Ecuador CIF-CFM-001.001 Chancellor Grant 2016 Universidad San Francisco de Quito and Microbiology Institute Contrato Marco de Acceso a los Recursos Genéticos MAE-DNB-CM-2016-0046 This work is supported by funding from the Proyectos Semilla from Universidad Central del Ecuador grant CIF-CFM-001.001, Chancellor Grant 2016 and Mini-Grant from Universidad San Francisco de Quito and Microbiology Institute, respectively; under Contrato Marco de Acceso a los Recursos Genéticos No. MAE-DNB-CM-2016-0046. The funders had no role in study design, data collection and analysis, decision to publish, or preparation of the manuscript.

==============================
Background

Bacterial vaginosis (BV) is a microbial imbalance (i.e., dysbiosis) that can produce serious medical effects in women at childbearing age. Little is known, however, about the incidence of BV or vaginal microbiota dysbiosis in pregnant teenagers in low and middle-income countries such as Ecuador. The scope of this exploratory analysis was to study the relationship between epidemiologic and microbial risk factors. Among the microbiology risk factors this study investigated five Lactobacillus species, two of them know in preview studies as microbiology risk factors for BV development (Lactobacillus acidophilus and Lactobacillus iners), and the last three known for being associated with a healthy vaginal tract (Lactobacillus crispatus, Lactobacillus gasseri and Lactobacillus jensenii). In addition, fastidious anaerobes known to be microbial risk factors for BV development in pregnant teenagers were searched as well, more exactly, Gardnerella vaginalis, Atopobium vaginae and Mobiluncus mulieris.

Methods

Ninety-five healthy adolescent pregnant women, visiting a secondary level hospital in Quito, Ecuador, were enrolled into the study in 2015. The enrolled patients were between 10 to 13 weeks of pregnancy. Four epidemiological risk factors were collected in a survey: age, civil status, sexual partners and condom use. Also, vaginal pH was measured as a health risk factor. DNA was extracted from endocervical and exocervical epithelia from all the patients’ samples. PCR analysis was performed in order to characterize the presence of the eight bacterial species known as risk factors for BV development, targeting three anaerobes and five Lactobacillus species. Univariate and multivariate analysis were performed to identify associated factors for the presence of anaerobic species using logistic regression.

Results

The 95 vaginal microflora samples of these teenagers were analyzed. Two of the bacterial species known to cause BV: A. vaginae (100%) and G. vaginalis (93.7%) were found in high prevalence. Moreover, the most predominant bacterial Lactobacillus species found in the pregnant teenagers’ vaginal tract were L. crispatus (92.6%), L. iners (89.5%) and L. acidophilus (87.4%). In addition, the average vaginal pH measured in the study population was 5.2, and high pH was associated with the presence of the three-anaerobic species (p = 0.001). Finally, L. jensenii’s presence in the study decreased in 72% the occupation of the three anaerobes.

Discussion

This work identified a high pH as a risk factor for BV anaerobes’ presence in adolescent pregnant women. Moreover, this study identified L. crispatus, L. iners and L. acidophilus to be the most abundant species in our study population. From all fastidious anaerobes analyzed in this study, A. vaginae was present in all pregnant teenagers. To conclude, L. jensenii could be a potential healthy vaginal microbiota candidate in pregnant teenagers and should be further analyzed in future studies.

Introduction

The bacterial microbiota of the vagina includes a diverse set of species (Ma, Forney & Ravel, 2012), consisting of a balance between anaerobic and aerobic microorganisms (Alvarez-Olmos et al., 2004). Commensal microorganisms in the vagina (vaginal microbiome) provide protection against opportunistic and pathogenic bacteria, constituting the first line of defense against invasive microorganisms (Ma, Forney & Ravel, 2012; Romero et al., 2014).

Several species of Lactobacillus sp. are dominant members of the vaginal commensal barrier and are usually associated with a healthy mucous membrane. Some of the species such as L. crispatus, L. gasseri, L. jensenii among others contribute in lowering pH through the production of great amounts of lactic acid and bactericide compounds (Wilks et al., 2004; Husain et al., 2014), such as H2O2 and bacteriocins (Wilks et al., 2004; Kiss et al., 2007; Tamrakar et al., 2007; Nelson et al., 2009; Machado et al., 2013; O’Hanlon, Moench & Cone, 2013; Romero et al., 2014). All these compounds and properties allow vaginal epithelial homeostasis and also keeping away possible pathogenic bacteria. However, not all Lactobacillus species found in the vaginal tract are strong probiotic species because of their low bactericide elements production such as L. acidophilus (Wilks et al., 2004), L. iners (Tamrakar et al., 2007) and others.

The vaginal microbiota of healthy women, during their reproductive age, is typically stable (Romero et al., 2014). When a woman is pregnant, however, hormone levels fluctuate, causing changes in the vaginal microbiota (Huang et al., 2014; Romero et al., 2014; MacIntyre et al., 2015). This fluctuation can be even more dramatic in teenagers, who may be at an increased risk for sexually transmitted diseases (STDs) and bacterial vaginosis (BV) (Alvarez-Olmos et al., 2004). In Ecuador, teenagers have a wide range of health care needs, in particular related to sexual and reproductive health, where the high rates of adolescent pregnancy and its health control are a national concern (Goicolea, 2010; Loaiza & Liang, 2013; Svanemyr et al., 2017).

BV is a condition in which the vaginal microbiota suffers a shift associated with a reduction in several species of probiotic Lactobacillus and an increase in the presence of anaerobes (G. vaginalis, A. vaginae and Mobiluncus sp.). BV also causes an individual to be more susceptible to STDs or suffer pre-term labor and late miscarriage (Wilks et al., 2004; Tamrakar et al., 2007; Nelson et al., 2009; Romero et al., 2014; McMillan et al., 2015).

The vaginal microbiota in pregnant teenagers has been characterized in certain studies, however, these studies are often conducted in high-income countries where hygiene conditions differ from low-income countries (Kiss et al., 2007; Nelson et al., 2009; Verstraelen et al., 2009; Aagaard et al., 2012; Huang et al., 2014; Husain et al., 2014; Hyman et al., 2014; Romero et al., 2014; MacIntyre et al., 2015). There are few studies that have been conducted in low- and middle-income countries (LMICs) such as Ecuador (Vaca et al., 2010) or Brazil (Ferreira et al., 2015), yet those analyses relied only on Nugent scores (gram stained smears). This study was therefore conducted to assess the presence of five Lactobacillus sp. (L. crispatus, L. gasseri, L. jensenii, L. acidophilus and L. iners) and three bacterial anaerobes (A. vaginae, G. vaginalis and M. mulieris) associated with BV in pregnant teenagers from Ecuador using PCR amplification of 16S and 23S rRNA genes.

Materials and Methods

Study area, design and subject selection

The study was carried out in a secondary level public hospital in Quito, Ecuador that serves as a national reference hospital and teaching hospital, which provides care to pregnant women, including family planning and postpartum support services (Moya, 2001). The hospital also provides services for pregnant teenagers (most of them from Hispanic ethnicity), and in 2014, 2,230 pregnant teens received assistance (Ministerio de Salud Pública del Ecuador, 2014).

From September 2015 to November 2015, 95 pregnant teenage women (10–19 years old) were enrolled into the study. The study population involved teenagers between 10 to 13 weeks pregnant and reported to have not taken antimicrobials for the previous 3 months. The enrolled patients were interviewed in a private room and survey included demographic and health questions, including: age, marital status, number of sexual partners in the last year and the use of preservatives (condoms) during intercourse. Applicants were excluded from the study if they reported having sexual intercourse within the last 48 h, or in case there was any evidence of macroscopic cervical bleeding or placenta and fetal disease. This investigation adopted a cross-sectional study design to determine the association between the presence of Lactobacillus sp. and BV related anaerobes.

Ethics statement

Ninety-five volunteers met the eligibility criteria and read and signed the informed consent (if 18 or older) or had their parents or legal representative sign if they were less of 18 years old. The Ethics Committee of the Central University of Ecuador approved this study (Protocol code: cif-cv-fcm.1).

Samples collection

Vaginal samples were collected by trained gynecologists. The procedure began by removing the cervical mucus with a sterile swab. Furthermore, endocervical and exocervical epithelia were gathered with a cervix examination brush (Rovers Cervex-Brush; Rovers Medical Devices B.V., Oss, The Netherlands). Additionally, the pH of the mucus samples was determined from the vaginal swab using a pH test strip (MColorpHast; Merck-Millipore, Burlington, MA, USA). The detachable head of the cervix examination brush was preserved in a liquid vial (BD SurePath™ liquid-based Pap test; BD Biosciences, San Jose, CA, USA) and transported under refrigeration conditions (4 °C). The samples were stored at −40 °C for 5 h until DNA extraction.

DNA extraction

DNA from the vaginal brush was extracted by following commercial kit instructions (NucleoSpin® Tissue 740952; Macherey-Nagel™, Düren, Germany) (Macherey-Nagel, 2014). Briefly, the brush samples were placed in 1.5 ml microtubes with 180 µl Buffer T1 and 25 µl Proteinase K. Then, to homogenate the mixture, the microtubes were centrifuged for 15 s at 1,500 × g and heated at room temperature for 5 min. Next, the tubes were vortexed vigorously for 15 s and span for 15 s at 1,500 × g. The microtubes were then incubated for 10 min at 70 °C, and followed by 5 min at 95 °C. The tubes were centrifuged again for 15 s at 1,500 × g, after which 200 µl of Buffer B3 was added. Tubes were vortexed vigorously for 30 s and heated at 70 °C for 10 min. Ethanol (96–100%) was added to each sample and vigorously vortexed once more. A Nucleospin Tissue Column was placed into a collection tube and the sample was added to the column. The tubes were then centrifuged for 1min at 11,000 × g and discarded and new ones were placed under the column. Kit buffers (BW and B5) were added to the columns and centrifuged for 1min at 11,000 × g for each buffer. The columns were dried with another centrifugation and finally DNA was eluted with buffer (BE) (Macherey-Nagel, 2014). The concentration of the extracted DNA was measured in a NANOVUE spectrophotometer (GE Healthcare Life Sciences, Little Chalfont, UK) to determine the DNA extraction quality. The original DNA extraction was then divided into two diluted aliquots with Buffer BE to a final concentration of 20 ng/µL. Finally, the remaining original samples were preserved at −80 °C, as well as the two aliquots of 20 ng/µL, were preserved at −20 °C for Polymerase Chain Reaction (PCR) analysis.

Polymerase chain reaction

PCR amplification was performed for nine different primer sets, targeting three anaerobes and five Lactobacillus species and one primer set for Lactobacillus genus of vaginal microbiota (see Table 1). Samples of L. jensenii and A. vaginae were sequenced to confirm their identity due to the lack of a strong specificity, while L. acidophilus, G. vaginalis and M. mulieris primer sets showed a strong specificity and thus samples did not require a further sequencing confirmation. The eventual confirmation step used the following universal primers for 16S rRNA sequencing (27Fw-AGA GTT TGA TCM TGG CTC AG and 805Rw-GAC TAC CAG GGT ATC TAA TC; temperature of annealing: 62 °C; Tanner et al., 1999) through a PCR assay carried out with a final volume of 50-µl (adapted from the procedure below) and sent to Functional Biosciences, Inc (Madison, WI, USA). The 16S rDNA sequences were compared to known sequences in GenBank with the advanced gapped BLAST (basic local alignment search tool) algorithm (Camacho et al., 2009).

Table 1 Primers used in this study.

Set	Name	Sequence (5′–3′)	Target	T (°C) of annealing	Size of fragment	Target	Specificity %	Cross reaction with microorganisms	Validation	Reference	
1	LactoF	TGGAAACAGRTGCTAATACCG	Lactobacillus spp.	55 °C	233 bp	16S rRNA	100%	Negative	Samples sequenced to confirm identity	Byun et al. (2004), Zhang et al. (2012)	
LactoR	GTCCATTGTGGAAGATTCCC	
2	LinersF	GTCTGCCTTGAAGATCGG	L. iners	55 °C	158 bp	16S rRNA	100%	Negative	Samples sequenced to confirm identity	De Backer et al. (2007), Zhang et al. (2012)	
LinersR	ACAGTTGATAGGCATCATC	
3	LcrispatusF	TTACTTCGGTAATGACGTTA	L. crispatus	55 °C	966 bp	16S rRNA	100%	Negative	Samples sequenced to confirm identity	De Backer et al. (2007), Zhang et al. (2012)	
LcrispatusR	GGAACTTTGTATCTCTACAA	
4	LgasseriF	TCGAGCGAGCTTGCCTAGATGAA	L. gasseri	60 °C	372 bp	16S rRNA	100%	n/d	Samples sequenced to confirm identity	Zhang et al. (2012)	
LgasseriR	CGCGGCGTTGCTCCATCAGA	
5	LjenseniiF	AGTCGAGCGAGCTTGCCTATAGAAG	L. jensenii	57 °C	342 bp	16S rRNA	100%	Negative	Samples sequenced to confirm identity in this study	Garg et al. (2009), Zhang et al. (2012)	
LactoR	GTCCATTGTGGAAGATTCCC	
6	LA-1 Fw	TCAATCAAAGGAAGACGCAG	L. acidophilus	56 °C	221 bp	23S rRNA	100%	Negative	n/d	Tsai et al. (2010)	
LA-2 Rw	CGCTCGCAATTTCGCTTA	
7	Gard154-Fw	CTCTTGGAAACGGGTGGTAA	Gardnerella vaginalis	60 °C	301 bp	16S rRNA	100%	Negative	n/d	Henriques et al. (2012)	
Gard154-Rv	TTGCTCCCAATCAAAAGCGGT	
8	Mobil-577F	GCTCGTAGGTGGTTCGTCGC	Mobiluncus mulieris	62 °C	449 bp	16S rRNA	100%	Negative	n/d	Fredricks et al. (2007)	
M.mulie-1026R	CCACACCATCTCTGGCATG	
9	Atop109-Fw	GAGTAACACGTGGGCAACCT	Atopobium vaginae	62 °C	221 bp	16S rRNA	16.7%	5 from 6	Samples sequenced to confirm identity in this study	Henriques et al. (2012)	
Atop109-Rv	CCGTGTCTCAGTCCCAATCT	37.5%	15 from 24	
Notes.

N/d, non determined.

The PCR assays were performed with T100™ thermal cycler (Bio-Rad Laboratories, Hercules, CA, USA) in a reaction volume of 20 µL. The reaction mix included 4 µL 5X Green GoTaq® Flexi Buffer (Promega, Madison, WI, USA), 2 µL MgCl2 2.5 mM (Promega, Madison, WI, USA), 0.50 µL dNTP mix 10 mM (Promega, Madison, WI, USA), 0.20 µL GoTaq® Flexi DNA Polymerase (Promega, Madison, WI, USA), 1.0 µL of each primer 0.5 µM, 2 µL from each DNA sample and the remaining volume with molecular grade H2O.

PCR amplification first cycle consisted in a pre-melt phase at 94 °C for 2 min and then denaturation at 94 °C for 30 s. After that, annealing at each species temperature (see Table 1) was realized for 30 s, and extension was performed at 72 °C for 1 min. This was repeated for 29 more cycles to 31 for the bigger amplicons (>500 bp). An additional 5 min extension was included at the end of the cycles to complete the extension of the primers. The PCR products were visualized using electrophoresis in 2% agarose gels and staining with ethidium bromide 0.1%, with negative and positive controls provided by the Microbiology Institute at Universidad San Francisco de Quito.

Statistical analysis

Uni- and multivariate analysis were realized in order to identify risk factors such as: age; civil status; number of sexual partners; condom use; vaginal pH; and Lactobacillus spp. associated with the presence or absence of anaerobic species (i.e., outcome) using logistic regression. Most variables were treated as categorical. For testing differences in the studied factors and vaginal microbiota between the groups of patients, the study was defined by the presence of three anaerobic species (A. vaginae, G. vaginalis and M. mulieris). Chi-square statistics were computed and logistic regression was used to estimate odds ratios for association between studied factors on the presence of three anaerobic species (A. vaginae, G. vaginalis and M. mulieris). Statistically significant differences were assumed when P-values were equal or less than 0.05. Statistical analyses were performed using STATA version 14.0 (StataCorp, College Station, TX, USA).

Results

Characteristics of study population

Most of the study participants were between 16–17 years old (53.7%) and non-married (68.4%). Approximately two thirds had only one sexual partner (76.8%) and around half did not use condoms (52.6%). In addition, all enrolled participants were from Hispanic ethnicity and nearly one third of the study population had a vaginal pH higher than five (31.5%). The pH mean value found in this study group was of 5.2 with a standard deviation (SD) of 0.7 and an interval of 4–7, Table 2 provides descriptive statistics of the study population.

Table 2 Epidemiological characteristics of pregnant teenagers in this study.

Epidemiological data			
	N°	%	
Age			
12 to 15 years	18	19.0	
16 to 17 years	51	53.7	
18 to 19 years	26	27.4	
Civil status			
Single	65	68.4	
Free union	29	30.5	
Married	1	1.1	
Sexual partners			
One	73	76.8	
More than one	22	23.2	
Condom use			
No	50	52.6	
Sometimes (Occasionally, yes)	45	47.4	
Vaginal pH			
4 to 5	63	68.5	
6 to 7	29	31.5	

Commensal microbiota

All samples showed the presence of Lactobacillus spp. The most frequently detected were L. crispatus 88/95 (92.6%), L. iners 85/95 (89.5%) and L. acidophilus 83/95 (87.4%). While L. jensenii and L. gasseri were only found in 77/95 (81.1%) and 54/95 (56.8%) of the studied pregnant teenagers, respectively.

All study participants were colonized by at least two of the five Lactobacillus species, which were analyzed. The most frequent combination was L. crispatus and L. iners in 82.1%. L. crispatus and L. acidophilus, were the second most prevalent combination with 80.0%, respectively. Other types of combinations were less frequent: L. crispatus and L. jensenii (74.7%); L. crispatus and L. gasseri (55.8%); and L. crispatus, L. gasseri and L. jensenii (47.4%), as shown in Fig. 1.

Figure 1 Percentage of grouped Lactobacillus species found between the samples.

BV associated anaerobes

The main BV associated anaerobic bacteria found in our study population was A. vaginae (found in 100% of the samples) followed by G. vaginalis and M. mulieris (in 93.7% and 34.7% of the samples, respectively).

When we looked at the presence of anaerobic bacteria in combination with the presence of Lactobacillus species, less than one-fifth of the samples had five Lactobacillus spp. and the three-anaerobic species (17.5%). Nevertheless, as showed in Fig. 2, no statistical significance was found for the association between the number of Lactobacillus species and the presence of three anaerobic species (P-value for Fisher’s exact test >0.05).

Figure 2 Percentage of teenagers with and without the presence of the three BV-associated anaerobes by the number of Lactobacillus species pre-existent in the vaginal epithelium.

Presence or absence of three anaerobic species by number of Lactobacillus species.

Factors associated with presence of Gardnerella vaginalis, Atopobium vaginae and Mobiluncus mulieris

In univariate analysis, higher pH was statistically associated with the presence of three anaerobes. Having a pH higher than 5 increased the likelihood that three anaerobes would be present in the samples (crude OR 4.7, 95% CI [1.82–12.3]), as shown in Table 3. In contrast, the presence of L. jensenii in vaginal microbiota was associated with a decreased presence of three-anaerobic species (crude OR 0.28, 95% CI [0.1–0.81]), as shown in Table 4.

Table 3 Factors associated with the presence or absence of three anaerobic bacterial species (Gardnerella vaginalis, Atopobium vaginae and Mobiluncus mulieris) that are known to be associated with bacterial vaginosis.

	Absent	Present	P-value	OR	P-value	
Age						
12 to 15 years	10 (15.4)	8 (26.7)		1		
16 to 17 years	34 (52.3)	17 (56.7)		0.62 (0.21–1.87)	0.400	
18 to 19 years	21 (32.3)	5 (16.7)	0.21	0.3 (0.08–1.14)	0.078	
Civil status						
Single	44 (67.7)	21 (70)		1		
Free union	20 (30.8)	9 (30)		0.94 (0.37–2.42)	0.900	
Married	1 (1.5)	0 (0.0)	0.99	–		
Condom use						
No	33 (50.8)	17 (56.7)		1		
Sometimes (occasionally, yes)	32 (49.2)	13 (43.3)	0.59	0.79 (0.33–1.88)	0.593	
Sexual partners						
One	48 (73.9)	25 (83.3)		1		
More than one	17 (26.1)	5 (16.7)	0.31	0.56 (0.19–1.71)	0.312	
pH						
4 to 5	50 (79.4)	13 (44.8)		1		
6 to 7	13 (20.6)	16 (55.2)	0.001	4.7 (1.82–12.3)	0.001	

Table 4 Association of Lactobacillus spp. with the presence or absence of three anaerobic bacterial species (Gardnerella vaginalis, Atopobium vaginae and Mobiluncus mulieris) that are known to be associated with bacterial vaginosis.

Bacterial species	Absent	Present	P-value	OR	P-value	
Lactobacillus spp.						
Absence	(0.0)	0 (0.0)		1		
Presence	65 (100.0)	30 (100.0)		does not apply		
Lactobacillus iners						
Absence	6 (9.2)	4 (13.3)		1		
Presence	59 (90.8)	26 (86.7)	0.54	0.66 (0.17–2.5)	0.547	
Lactobacillus crispatus						
Absence	4 (6.1)	3 (10)		1		
Presence	61 (93.9)	27 (90)	0.51	0.59 (0.12–2.8)	0.509	
Lactobacillus gasseri						
Absence	25 (38.5)	16 (53.3)		1		
Presence	40 (61.5)	14 (46.7)	0.17	0.55 (0.23–1.3)	0.176	
Lactobacillus jensenii						
Absence	8 (12.3)	10 (33.3)		1		
Presence	57 (87.7)	20 (66.6)	0.01	0.28 (0.1–0.81)	0.02	
Lactobacillus acidophilus						
Absence	6 (9.2)	6 (20)		1		
Presence	59 (90.8)	24 (80)	0.14	0.41 (0.12–1.38)	0.151	
Notes.

women/state; Nd, Not detected.

When the associations (Table 3) were adjusted for age, condom use and number of sexual partners, the only significant factor associated with presence of three anaerobes was a high pH (adjusted OR = 5.5; 95% CI [2–15.2], P-value: 0.002). After adjusting for the same variables in Table 4, the presence of L. jensenii remained to be associated with the absence of three anaerobes (adjusted OR 0.16; 95% CI [0.04–0.6], P-value: 0.006).

Discussion

In the current study, we found that the genital tract of pregnant Ecuadorian teenagers was colonized primarily by L. crispatus. All samples from the studied teenagers were positive for at least one analyzed Lactobacillus species. Additionally, from the five-species analyzed in this study, colonization by the consortium formed through L. crispatus, L. iners and L. acidophilus was the most prevalent. These results coincide with previous reports indicating that Lactobacillus species is the most abundant commensal bacteria found in healthy vaginal microbiota of pregnant and non-pregnant adult women in the world, as shown in Table 5 (Kiss et al., 2007; Tamrakar et al., 2007; Verstraelen et al., 2009; Dominguez-Bello et al., 2010; Hernández-Rodríguez et al., 2011; Aagaard et al., 2012; Husain et al., 2014; Hyman et al., 2014; Romero et al., 2014; MacIntyre et al., 2015; McMillan et al., 2015). However, it is important to mention that other Lactobacillus species could have been present on the vaginal epithelium of these pregnant teenagers because this study did not analyze every Lactobacillus species but only the most frequently cited in other studies (see Table 5). We found a large proportion of individuals colonized by BV associated anaerobic bacteria, including: A. vaginae (100%), and G. vaginalis (93.7%), which showed some discrepancies with previous studies carried out in adult women (Table 5). The percentage of A. vaginae in this study was considerably different from other published studies due to the presence of A. vaginae in 100% of the analyzed pregnant teenagers in Ecuador. In agreement with the previous results, G. vaginalis was also found in more than 93% of the pregnant teenagers. As referenced by previous studies, G. vaginalis and A. vaginae are the main anaerobes associated with BV (Menard et al., 2008; Verstraelen & Swidsinski, 2013; Bretelle et al., 2015; Machado & Cerca, 2015; Hardy et al., 2016).

Table 5 Summary of vaginal microbiota characterization studies in pregnant women (including this study).

N°	Population description	Study group (n)	Place	Methodology	pH (mean)	Bacterial species detected (%)	Author	
						Lactobacillus sp.	L. iners	L. crispatus	L. acidophilus	L. jensenii	L. gasseri	G. vaginalis	A. vaginae	M. mulieris	Other Lactobacillus sp.		
1	Pregnant women (Age range 18–35)	126	Austria	Multiplex PCR by Culture	Nd	57.1	Nd	19.8	Nd	11.9	Nd	Nd	Nd	Nd	YES	Kiss et al. (2007)	
2	Pregnant women (Age range Nd)	100	Belgium	tRFLP—PCR	Nd	X	40.3	23.4	Nd	3.9	40.3	Nd	Nd	Nd	NO	Verstraelen et al. (2009)	
3	Pregnant and non-pregnant women (Age range 19.4–39.2)	34	China	PCR and 16 sregion sequencing	4.5	X	55	26.9	Nd	Nd	6.3	11.8	11.8	Nd	NO	Huang et al. (2014)	
4	Pregnant teenagers (Age range 12–19)	95	Ecuador	PCR
Electrophoresis gel	5.2	100	89.5	92.6	87,4	48.4	56.8	93.7	100	34.7	NO	This study (2018)	
5	Pregnant women (Age range 19–44)	132	Japan	PCR
Culture	Nd	98.5	41.7	51.5	Nd	25	31.8	Nd	Nd	Nd	NO	Tamrakar et al. (2007)	
6	Pregnant women (Age range 13–43)	140	Mexico	PCR (DGGE)	Nd	98.4	56.7	Nd	78.1	Nd	20.3	Nd	Nd	2.0	YES	Hernández-Rodríguez et al. (2011)	
7	Pregnant and non-pregnant women (Age range 18–55)	131	Rwanda	16s region pyrosequencing	4.6	X	X	X	Nd	X	X	X	X	Nd	NO	McMillan et al. (2015)	
8	Pregnant women (Age range Nd)	42	United Kingdom	16s region pyrosequencing	Nd	X	30a	43a	Nd	14a	9a	Nd	2a	Nd	YES	MacIntyre et al. (2015)	
9	Pregnant women (Age range 28–31)	293	United Kingdom	Culture
MALDI-TOF	Nd	75	Nd	37	Nd	45	34	Nd	Nd	Nd	YES	Husain et al. (2014)	
10	Pregnant and non-pregnant women (Age range Nd)	54	USA	16s region pyrosequencing	Nd	X	58.5a	38.1a	Nd	X	4.3a	2.2a	2.2a	Nd	YES	Romero et al. (2014)	
11	Pregnant and non-pregnant women (Age mean 31.4)	84	USA	16s region pyrosequencing	Nd	X	X	X	Nd	X	Nd	Nd	Nd	Nd	YES	Aagaard et al. (2012)	
12	Pregnant women (Age ≥18)	88	USA	PCR
Sanger sequencing	Nd	X	76.9 African
21.7 Asian
23.1 Caucasian
68.4 Hispanic	30.8 African
41.5 Caucasian
15.8 Hispanic	21.7 Asian
1.5 Caucasian	7.7 African
27.7 Caucasian
31.6 Hispanic	7.7 African
17.4 Asian
41.5 Caucasian
10.5 Hispanic	Nd	X	Nd	YES	Hyman et al. (2014)	
13	Pregnant women (Age mean 24)	50	USA	qPCR	Nd	Nd	Nd	X	Nd	Nd	Nd	66.7	Nd	Nd	NO	Nelson et al. (2009)	
14	Pregnant women (Age range 21–33)	9	Venezuela	PCR
yrosequencing	Nd	88–94	Nd	Nd	Nd	Nd	Nd	0.1–2	4–34	Nd	NO	Dominguez-Bello et al. (2010)	
Notes.

Legend: X, species analyzed but no percentage reported; Nd, Not detected.

a Results only from pregnant women/state.

The association between having a certain commensal vaginal microbiota with a healthy pregnancy outcome has been thoroughly studied in the past two decades (Kiss et al., 2007; Nelson et al., 2009; Verstraelen & Swidsinski, 2013). It has been postulated that an optimal commensal microbiota in the vaginal epithelium reduces the risk of infections with any pathogen (Trichomonas vaginalis, Neisseria gonorrhoeae, Chlamydia trachomatis, Candida sp., and others) in the reproductive track and also seems to prevent preterm birth caused by bacterial vaginosis (O’Hanlon, Moench & Cone, 2013; Krauss-Silva et al., 2014). Several studies around the world (see Table 5), such as in USA (Hyman et al., 2014; Romero et al., 2014), United Kingdom (MacIntyre et al., 2015), Belgium (Verstraelen et al., 2009), China (Huang et al., 2014) and Japan (Tamrakar et al., 2007), showed that the most abundant Lactobacillus species found in pregnant women are L. iners and L. crispatus. In agreement to this study, these two species were the most abundant lactobacilli found in pregnant teenagers in Ecuador. The third most abundant Lactobacillus species found in this study was L. acidophilus, which has only been observed in studies from Mexico (Hernández-Rodríguez et al., 2011) and USA (Hyman et al., 2014). In USA, it has been encountered in non-pregnant teenager’s vaginal microbiota with a prevalence of 49% (Alvarez-Olmos et al., 2004). L. acidophilus has not been analyzed or encountered in other studies worldwide as shown in Table 5.

Other Lactobacillus species, such as L. jensenii and L. gasseri, are usually found in high percentages in other studies of vaginal microbiota present in pregnant women (Tamrakar et al., 2007; Verstraelen et al., 2009; Husain et al., 2014; Hyman et al., 2014; MacIntyre et al., 2015). However, Lactobacillus species, such as L. delbruecki, L. rhamosus, L. reuteri, L. casei. L. paracasei, L. vaginalis, L. coleohominis, L. fermentum, L. fornicalis, L. gallinarum, L. helveticus, L. kefiranofaciens, L. kitasatonis, and L. ultunensis, have been analyzed as well but are not commonly found in studies of pregnant women’s vaginal microbiota (Hyman et al., 2014). Nonetheless, L. jhonsonii was detected by 16S rRNA pyrosequencing characterization of vaginal microbiota in pregnant women in USA (Aagaard et al., 2012), although this species was not analyzed in the current study. Therefore, a more extensive analysis may be necessary in the future.

Atopobium vaginae was the most frequent anaerobe observed in vaginal microbiota from pregnant teenagers in our study (Table 5), and it has been detected in USA (Hyman et al., 2014; Romero et al., 2014), Mexico (Hernández-Rodríguez et al., 2011), China (Huang et al., 2014), Europe (Bretelle et al., 2015; MacIntyre et al., 2015) and Africa (McMillan et al., 2015). This anaerobe is usually found in association with G. vaginalis and M. mulieris (Hernández-Rodríguez et al., 2011) in the development of BV (Verstraelen & Swidsinski, 2013). As similar as G. vaginalis, A. vaginae is usually found in minor percentages around the world (Nelson et al., 2009; Dominguez-Bello et al., 2010; Huang et al., 2014; Romero et al., 2014; MacIntyre et al., 2015). The percentage of A. vaginae in this study was markedly different from other published studies. In our study, A. vaginae was present in 100% of the analyzed pregnant teenagers in Ecuador. In agreement with previous results, G. vaginalis was also found in more than 93% of pregnant teenagers. Although this study did not quantify the abundance of G. vaginalis or A. vaginae, the prevalence of these anaerobes is not concordant with the normal prevalence in healthy pregnant women in other studies (Nelson et al., 2009; Dominguez-Bello et al., 2010; Huang et al., 2014; Romero et al., 2014), such as Africa where 66.7% of healthy pregnant women had G. vaginalis (Nelson et al., 2009) or the USA where 2.2% of healthy pregnant women had G. vaginalis or A. vaginae (Romero et al., 2014), which is low in comparison with the results from this study. Finally, only 34.5% of the pregnant teenagers were positive for M. mulieris, which is high in comparison to other studies performed in pregnant women such as 2.0% in Mexico (Hernández-Rodríguez et al., 2011), 0.37% in Rwanda (Tchelougou et al., 2013) and 18.2% in USA (Waters et al., 2008).

The major drawback of this study was the lack of quantitative data which may allow us to assess the status of the colonization of the distinct bacterial taxa. However, BV is a very common cause for high pH in the vagina, associated with the shifting of microbiota from protective Lactobacillus sp. to anaerobes, such as G. vaginalis and A. vaginae (De Backer et al., 2007). When this occurs, the pH rises from 4.2 that is considered normal to 4.5 or higher, which is now used as a positive indicator for BV (O’Hanlon, Moench & Cone, 2013). The pH in pregnant women has been analyzed in several studies, such as China (Huang et al., 2014) and Rwanda (McMillan et al., 2015) where the pH media was 4.5 and 4.6, respectively. In this study, however, the observed pH was 5.2, which is a marker for the shift in dominance from Lactobacillus to anaerobic microbiota on the vaginal epithelium. The presence of the three anaerobes analyzed in this study appeared to increase the vaginal pH, which is in agreement with other studies (McMillan et al., 2015; Donders et al., 2016) that evaluated the establishment of BV in women.

This exploratory study was able to characterize the heterogeneity of the commensal Lactobacillus species in these pregnant teenagers, where all study participants had more than one Lactobacillus species colonizing the vagina. As shown in Fig. 2, the reproductive track of our study participants revealed two to five Lactobacillus species. Even though there is no statistical significance with the absence or presence of the three anaerobes, our results differed from what was published in other studies (Hernández-Rodríguez et al., 2011; Hyman et al., 2014). In Mexico, it has been reported that the vagina was colonized by one to four species (Hernández-Rodríguez et al., 2011), meanwhile in the USA by one to two species (Hyman et al., 2014). L. crispatus, L. gasseri and L. jensenii (Wilks et al., 2004; Husain et al., 2014) are known as strong beneficial species, however L. acidophilus (Wilks et al., 2004), and L. iners (Tamrakar et al., 2007) are known as weaker protective species. Therefore, the high prevalence of these two species in the vaginal microbiota could partially explain the high prevalence of A. vaginae and G. vaginalis in Ecuadorian pregnant teenagers. Even without a statistical difference, the high prevalence of L. iners and L. acidophilus could imply some sort of bewilderment in the commensal bacteria of the vaginal microbiota, which could allow the proliferation of anaerobic bacteria. Further study is necessary to quantify each Lactobacillus species and anaerobic bacteria in each sample in order to determine the exact load of the detected vaginal microbiota in pregnant teenagers. Also, the amount of the beneficial substances produced by Lactobacillus species found in Ecuadorian women, such as H2O2 or lactic acid, should be quantified to determine their protective ability.

Finally, this study limitations were principally caused by the low number of samples, and the lack of bacterial abundance detection.

Conclusion

Our study identified L. crispatus, L. iners and L. acidophilus to be the most abundant species in our study population of pregnant teenagers. L. jensenii could be a potential microbiota protective candidate in the vaginal microbiota in pregnant Ecuadorian teenagers. Meanwhile, A. vaginae, G. vaginalis and M. mulieris were found in 100%, 93.7% and 34.7% of the analyzed pregnant teenagers, respectively.

To the authors’ knowledge, this is the first study of vaginal microbiota in pregnant teenagers in Ecuador. Furthermore, this investigation was only realized in pregnant teenagers and there was no control group (non-pregnant teenagers). Further studies are necessary in pregnant women to quantify the main Lactobacillus species and also the BV associated anaerobes identified in this pilot study, as well as to statistically determine a possible correlation with preterm birth or BV in Ecuadorian pregnant women, which was not possible to characterize in this exploratory study.

Supplemental Information

File S1 Raw data epidemiological file from this study

Click here for additional data file.

We thank Jay Graham, Gabriel Trueba and Sonia Zapata for their valuable contributions to the manuscript, the volunteers who made this study possible, and also the people who work in the Institute of Microbiology of USFQ (in particular Dario Cueva, who helped in the validation of L. jensenii primers) and the staff from the Biomedicine Centre of the Central University for their help and support.

List of abbreviations

BV Bacterial vaginosis

STDs Sexually Transmitted Diseases

LMICs Low and Middle-Income Countries

Additional Information and Declarations

Competing Interests

Author Contributions

Human Ethics

Ethics

Data Availability

The authors declare there are no competing interests.

Ana María Salinas performed the experiments, analyzed the data, wrote the paper, prepared figures and/or tables, reviewed drafts of the paper.

Verónica Gabriela Osorio performed the experiments.

Pablo Francisco Endara analyzed the data, contributed reagents/materials/analysis tools, wrote the paper.

Eduardo Ramiro Salazar and Gabriela Piedad Vasco contributed reagents/materials/analysis tools, reviewed drafts of the paper, clinical samples.

Sandra Guadalupe Vivero performed the experiments, contributed reagents/materials/analysis tools, clinical samples.

Antonio Machado conceived and designed the experiments, analyzed the data, wrote the paper, reviewed drafts of the paper.

The following information was supplied relating to ethical approvals (i.e., approving body and any reference numbers):

The Ethics Committee of the Central University of Ecuador approved the study (Protocol code: cif-cv-fcm.1).

The following information was supplied relating to ethical approvals (i.e., approving body and any reference numbers):

The Institutional Review Board of the USFQ (Ref.: MSP-VGVS-2016-0244-O).

The following information was supplied regarding data availability:

The raw data has been provided as a Supplemental File.

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
