# Peer review of "Bacterial identification of the vaginal microbiota in Ecuadorian pregnant teenagers: an exploratory analysis"

_PeerJ, doi:10.7717/peerj.4317_

## Round 0.1 · original submission · Major Revisions

Dear Authors,

The Reviewers found your manuscript very interesting, recommending a thorough revision in order to achieve publication.

I would suggest to take into consideration the Reviewers' comments, discuss and incorporate them within your manuscript in order to reach the standard requested for publication.

Best regards

Salvatore Andrea Mastrolia
PeerJ Academic Editor

·

Basic reporting

Summary: The authors reported the valuation of eight vaginal bacterial species, known to be risk or protective factors for bacterial vaginosis, and the pH value in a specific cohort: pregnant teenagers in Ecuador. Surely, considering the correlation between dysbiosis and some adverse pregnancy outcomes, this study could implement our knowledge on vaginal flora, also in low- and middle-income countries. A limit on the study, reported by the authors, was the absence of a control study and, another one, was the lack of information on pregnancy outcomes in this population.

Basic reporting
The English language was intelligible, but errors, repetition and long sentence were present in the text.
The literature available was summarized in the test. Considering this is an observational study, more information could be added about data reported in other studies to give a point on the topic of the study.
Figure are of high quality, but I think that the graphs do not add any additional information and the text could be enough.
Table are clear; I have only to underlined that table 2,3 and 4 are all named as table 1.

Experimental design

See “general comment”

Validity of the findings

This study could give a description on partial vaginal microbiota in pregnant teenagers in Ecuador. The novelty of the study correlated with the population take into consideration. The limit of results correlated with the little simple size (as reported by the authors) and the absence of a consistent clinical impact of the data found. In fact, no all bacterial species were searched, and no correlation on maternal and fetal outcomes were reported. Even if it is not possible to define an influence on adverse outcome, as explained by the authors, a description of incidence of major complications of pregnancy in the population may be of interest.

Additional comments

English language required some improvements. For example:
- frequent repetition of word is used. See rows 45-46; rows 127-140
- A long sentence was reported at row 61-66. Please summarized it or try to clearly explain the concept.
Specific comments
Title: The title is appropriate.
Summary/ Abstract: Some inaccuracies were reported. Please clarify the sequent points.
- The author reported that “five risk factors were collected”, but only four were listed [rows 31-32]
- They also reported that nine bacterial species were searched, but only 8 were defined [row 34]
Introduction: The literature available was summarized in the test. Considering this is an observational study, more information could be added about data reported in other studies to give a point on the topic of the study.

Materials and methods: 1) how risk factor analyzed in the test were choose?
2) Are there any information of previous use of hormonal contraceptive? In could be of interest to verify if this factor, associated with change in vaginal pH, could be correlated with BV
3) Considering the fact that the patients were pregnant, how the use of condom was defined? Use of it before pregnancy research? Use during pregnancy? Please clarify
4) Why placental and fetal disease were excluded?
5) Considering that vaginosis is a risk factor, but it is not always associated with adverse obstetric outcomes, could the authors reported the pregnancy course or their population? In fact, even they reported the association of VB and preterm delivery in other studies, could be of interest to value this condition in the particular population collected in the study (Ecuadorian pregnant teenagers).

Discussion: The discussion is too short and it should extended in a systematic manner. List the principal findings and then discuss them in comparison with other literature, reporting more detailed on novelty of the study, why some risk factor were analyzed while other were not take into consideration.
Tables: The material provided by the authors is clear. Table 2, 3 and 4 are called table 1

·

Basic reporting

Numbers of pages are not indicated

Introduction is clear and complete, the available literature is well illustrated in the text.

English language used is correct, but some errors are present in the text, an example: line 198: error: “All study participants were colonized with at least by two of the five Lactobacillus”.

Tables and figures are clear and well constructed. But table 2, 3 and 4 are all entitled as table 1

Experimental design

The study is just observational and the conclusions, in line with the aim declared: a characterization of bacterial and Lactobacillus species present in Ecuadorian pregnant teenagers, are merely descriptive.

In the study there is no mention of patients’ symptoms. Would have been interesting register how the presence or absence of certain species of bacteria or Lactobacillus correlated with symptoms declared by the patients.

Even if the authors themselves reported the known association between presence or absence in vaginal tract of pregnant women of certain microbiota species and the risk of preterm labor and colonization from pathogens, they did not investigate this point in their study.

Discussion should be improved. I suggest to list the findings progressively comparing them with available literature in order to illustrate the innovation bring from the study.

Validity of the findings

Relation between high pH and BV was already known and is in accordance with previous literature; however this is the first study that investigate the population (even if not all species were taken into account) of lactobacillus sp. and bacterial anaerobes in a given specific population of Ecuadorian pregnant teenagers.

The novelty provided by the authors is that the bacterial population found in this specific group is different from the ones reported in literature both in low and high income countries. The authors also found that L.Crispatus, L.iners and L. acidophilus are the species most present in their study population.

Results are consistent with declared aim of the study, but their clinical impact is scarce. Would have been more fruitful to investigate how these finding can help in management of BV in pregnancy, preventing development of BV and risk associated with this condition (such as colonization from other pathogens and preterm birth). For example recording the outcomes of pregnancy, describing in particular age at delivery and pregnancy complications.

Additional comments

I would suggest to perform different kind of studies in this particular population:
A comparison of bacterial colonization of vaginal tract between pregnant and non pregnant Ecuadorian teenagers; if the aim is just to characterize the bacterial population of vaginal tract.

Analyze how the presence of determined species correlates to pregnancy outcomes, such as vaginal colonization from pathogens and the risk of preterm birth

Reviewer 3 ·

Basic reporting

The manuscript entitled “Bacterial identification of the vaginal microbiota in Ecuadorian pregnant teenagers: An exploratory analysis” by Salinas et al., seeks to identify the presence of 5 Lactobacilli species and 3 BV associated anaerobes in the vagina of pregnant teenagers. The study is interesting, although there are several outstanding issues.

1) It is never clearly explained by the authors why this study targeted this population other than it hadn't been done before. Is there a high rate of BV in Ecuadorian teenagers? Is there a higher than expected rate of preterm delivery in Ecuadorian teenagers? Those would be valid reasons to pursue this research conditions, but there is no indication in the manuscript that is the case. Since the study involves the participation of a vulnerable population (pregnant minors), one would hope that there is an outstanding reason to pursue their involvement. The course and outcome of these pregnancies is also not provided, adding to the questions surrounding this study.
2) The English needs to be revised. There are major passages that are incomprehensible such as Line 32-33, and multiple odd grammar constructions throughout the manuscript.
3) There are multiple formatting issues, from inconsistent display of references (including/excluding year arbitrarily, display multiple authors at times, etc.).
4) Figures lack proper display. Figure is missing a Y-axis and error bars. Figure 2 is incomprehensible to me. I have no idea what is being displayed.
5) Table 1: One of headings is in Spanish.
6) The abstract should mention at what time point the subjects were sampled.
7) In the Methods it is mentioned that the samples were sequenced to confirm identity, yet no methodological details or results are provided.
8) Line 213: “…weak statistical evidence (P-value for Fisher’s exact test = 0.08)…” A P-value of 0.08 is not significant. Statistical significance either exists or does not exist. There is no weak or strong statistical evidence.
9) The ethnicity of the participants is not disclosed. This is an important parameter for the vaginal microbiome composition. The discussion should address this when comparing to other studies.
10) Line 254: “…this study did not analyze every Lactobacillus species only the most significant.” The selected species in this study were chosen because they were presumed to be the most common. Significance has nothing to do with this.

In summary, this study has multiple serious flaws.

Experimental design

Research question ambiguous. Missing data.

Validity of the findings

Primers were not validated with pure cultures prior to the experiment.

Additional comments

See above.

---

## Round 0.2 · Minor Revisions

Dear Authors,

I very much appreciated the efforts in addressing the Reviewer's comments. The Reviewers found your manuscript interesting and two of were favorable to publication.

However, Reviewer #3 raised some concerns and suggested publication to be considered only after an additional revision of the manuscript is achieved.

In light of the Reviewer's comments, I personally evaluated the manuscript and agree with the Reviewer's concerns.

I would suggest the authors to answer all comments within a rebuttal letter, discuss and incorporate them in the revised version of your manuscript in order to reach the standard requested for publication.

With personal regards

Salvatore Andrea Mastrolia, MD
PeerJ Academic Editor

·

Basic reporting

The English language and the structure of the text were implemented.
The literature available was summarized in the test and in the table.
Figure are of high quality; I already think that the graphs do not add any additional information and the text could be enough. My comment did not preclude the pubblications of this paper, but summarized the test could be suggested, if the authors would like to published also the graphs,
Table are clear

Experimental design

Original research
Aim and scope well clarify
Limits are present in the study, but the authors explain them satisfactorily
Investigation were conducted with good standards; ethical standard was respected and an informed consent was signed. Methods were clarely described

Validity of the findings

This study could give a description on partial vaginal microbiota in pregnant teenagers in Ecuador. The novelty of the study correlated with the population take into consideration. The limit of results are now well explained by the authors

Additional comments

I appreciate the improvements the authors have done to their manuscript
Even if limits persist in the study, they are well explained and the necessity of future evaluation is underlined
I think that the article meets the criteria of Peerj and it should be accepted now in the current form

·

Basic reporting

The author’s answer satisfy all my request.
They clearly explained the errors in numbers of pages and in table titles and corrected the English errors .
They improved the results session and the discussion.

Experimental design

The limit of this study remains (small size and experimental design), but, considering that this is a pilot study, the descriptive and observational nature of this work and the small sample size are acceptable.

Validity of the findings

As mentioned above this study has some limits and the results have scarce clinical impact, but this work is a pilot study. So, in view of setting up a bigger study with the aim of compare the bacterial flora in pregnant and non pregnant teenagers and of determine the correlation between BV and preterm birth, this could be a good start.

Additional comments

.

Reviewer 3 ·

Basic reporting

This manuscript is an improved version of the previous iteration. However, a few concerns remain. The English is improved but still contains multiple grammar issues. This version still needs to be improved.

Experimental design

Ok.

Validity of the findings

Results are valid but conclusions made are beyond the results presented and the scope of the study.

Additional comments

I continue to find the study design, presentation of results and conclusion somewhat disconnected. The reasoning for the study is reflected better in the rebuttal letter than in the manuscript for example. And while the presence of BV associated microorganisms can be seen as a sign of disease, it is not necessarily so. Many women carry these microorganisms without symptoms or deleterious effects. Given that pregnancy complications were not reported in this study, the statement that some of these adolescents were at increases risk simply for carrying these microorganisms is not supported if not accompanied by symptomatic manifestation or adverse pregnancy outcome. Given the complexity of interactions, carrying 3 anaerobes versus 1 of them does not necessarily equal a worse prognosis either, especially given the different interactions with the different Lactobacilli species. So, given the limited scope of the experimental testing, the analysis to support the conclusions would have to be much more extensive and be grounded on clinical outcome. In summary, either the study has to adopt a descriptive nature, meaning that it limits itself to presenting the data in terms of what is the most common bacteria found in this study population; or a much more elaborate analysis will have to be performed to support the current conclusions.

---

## Round 0.3 · accepted · Accept

Dear Authors,

I would like to compliment with you for the efforts provided in addressing the Reviewers' comments.

I feel that your manuscript has reached the level of publication and can be accepted in its current form.

Best regards

Salvatore Andrea Mastrolia
PeerJ Academic Editor